# Leveraging Blood-Based Diagnostics to Predict Tumor Biology and Extend the Application and Personalization of Radiotherapy in Liver Cancers

**DOI:** 10.3390/ijms23041926

**Published:** 2022-02-09

**Authors:** Franziska Hauth, Hannah J. Roberts, Theodore S. Hong, Dan G. Duda

**Affiliations:** 1Edwin L. Steele Laboratories for Tumor Biology, Massachusetts General Hospital, Boston, MA 02114, USA; FHAUTH@mgh.harvard.edu; 2Department of Radiation Oncology, Massachusetts General Hospital, Boston, MA 02114, USA; Hannah_Roberts@DFCI.HARVARD.EDU (H.J.R.); tshong1@mgh.harvard.edu (T.S.H.); 3Department of Radiation Oncology, University Clinic Tuebingen, 72076 Tuebingen, Germany

**Keywords:** circulating biomarkers, liver cancer, radiotherapy, toxicity, treatment personalization

## Abstract

While the incidence of primary liver cancers has been increasing worldwide over the last few decades, the mortality has remained consistently high. Most patients present with underlying liver disease and have limited treatment options. In recent years, radiotherapy has emerged as a promising approach for some patients; however, the risk of radiation induced liver disease (RILD) remains a limiting factor for some patients. Thus, the discovery and validation of biomarkers to measure treatment response and toxicity is critical to make progress in personalizing radiotherapy for liver cancers. While tissue biomarkers are optimal, hepatocellular carcinoma (HCC) is typically diagnosed radiographically, making tumor tissue not readily available. Alternatively, blood-based diagnostics may be a more practical option as blood draws are minimally invasive, widely availability and may be performed serially during treatment. Possible blood-based diagnostics include indocyanine green test, plasma or serum levels of HGF or cytokines, circulating blood cells and genomic biomarkers. The albumin–bilirubin (ALBI) score incorporates albumin and bilirubin to subdivide patients with well-compensated underlying liver dysfunction (Child–Pugh score A) into two distinct groups. This review provides an overview of the current knowledge on circulating biomarkers and blood-based scores in patients with malignant liver disease undergoing radiotherapy and outlines potential future directions.

## 1. Introduction

Liver malignancies are a leading cause of cancer-related death worldwide. Hepatocellular carcinoma (HCC) is the most common liver malignancy, and its incidence has more than tripled in the US since 1980 [1,2]. In recent years, radiotherapy has emerged as a promising modality for patients with liver malignancies, aided by advances in target immobilization, localization, and treatment delivery. Intensity modulated radiation therapy (IMRT) and volumetric arc therapy (VMAT) have allowed for highly conformal radiation plans while maximally sparing the surrounding healthy liver as much as possible. MRI-based online adaptive treatment is another promising technique to further spare healthy liver tissue [3,4,5]. Radiotherapy is generally a safe and highly effective treatment with local-control rates for HCC of up to 95% [6,7,8,9,10]. However, the optimal dose and fractionation scheme, as well as radiotherapy technique such as photon versus proton irradiation, remain unclear. In addition, the potential for severe toxicities such as radiation-induced liver disease (RILD) in patients with underlying liver diseases (cirrhosis, steatosis) may be dose-limiting in some patients. [11,12]. These challenges could be addressed by identifying accurate and meaningful biomarkers for liver cancer radiotherapy.

As new treatments emerge, the investigation of biomarkers and clinical scores to predict tumor biology is essential to guide patient management. Among several methods, blood-based diagnostics have unique advantages. Peripheral blood can be easily obtained and collected throughout the treatment and recovery periods, allowing for the measurement of toxicity, treatment response or disease progression. In addition, as blood draws are widely available without elaborate setups or imaging services, its use can be easily adopted into clinical practice. As virtual visits become increasingly common, blood-based biomarkers may facilitate remote monitoring without requiring patients to travel, potentially alleviating financial toxicities for patients undergoing cancer therapy [13]. The identification of circulating biomarkers in patients with HCC would be of particular use because the diagnosis is typically made radiographically; tissue biopsies are uncommon due to the risk of bleeding and needle-tract seeding [14]. This review provides an overview of current approaches on blood-based diagnostics for patients treated with radiotherapy for liver malignancies.

## 2. Clinical Scores: Child–Pugh, Barcelona Clinic Liver Cancer (BCLC), and Albumin–Bilirubin (ALBI) Grades

Historically, Child–Pugh score (CPS) system (also referred to as the Child–Turcotte–Pugh score) has been used to assess hepatic function in a variety of liver malignancies, by itself or as part of the Barcelona Clinic Liver Cancer (BCLC) [15]. Although widely used, there are major concerns on the objectivity of the CPS system and its ability to stratify patients into prognostic groups. First, the presence and severity of both ascites and encephalopathy are subjective, often assessed by a single physician. Second, both parameters are strongly influenced by extrinsic factors such as medication, making them prone to fluctuations that may affect the clinical grade. Finally, some variables are interdependent; for example, low albumin may lead to increased ascites, resulting in a potentially disproportional impact on the total score [16].

The BCLC staging system incorporates the CPS score as a measure of underlying hepatic function, along with the disease extent and Eastern Cooperative Oncology Group (ECOG) performance status to create five stages of HCC. These five stages range from very early stage (BCLC 0) to Terminal stage (BCLC D) and are predictive of prognosis. The BCLC score is widely used in clinical management.

Similarly, the albumin–bilirubin (ALBI) grade has been used to predict prognosis in HCC. Initially developed to assess of liver function after surgery [17,18], it was developed to be more objective than the CPS system. The ALBI grade uses plasma albumin and bilirubin using the formula ALBI = (log10 bilirubin [µmol/L] × 0.66) + (albumin [g/L] × −0.0852) to create the following three groups: grade 1: ≥−2.60, grade 2: −2.61–−1.39 and grade 3: >−1.39 [19]. Since first proposed in 2015 by Johnson and colleagues as a new and less biased approach towards assessment of liver function, there has been increasing interest in validating the ALBI score for liver cancer patients undergoing radiotherapy. Since 2015, more than 1000 patients have been included in studies of pre-treatment ALBI scores, predominantly using retrospective analyses (Table 1). In line with previously published data for patients undergoing liver resection or other liver directed therapies [20,21,22], several studies showed a strong correlation between overall survival and ALBI score [23,24,25,26,27]. However, the specific threshold of a statistically significant increase in ALBI score varied between the studies (e.g., Murray et al. [23]: 0.1 increase vs. Toesca et al. [25]: 0.5), making a cross-study comparison difficult. Of note, the ALBI score outperformed CPS system in predicting mortality in some cases [23,26,27,28].

Five studies showed a correlation between pre-treatment ALBI score and a decline in liver function following radiotherapy [23,24,25,28,29]. As previously described for other liver cancer therapies, the ALBI score was able to further subdivide patients in CPS group A into two distinct subgroups (ALBI grade 1 and 2) correlating both with overall survival and more importantly with the development of liver toxicity after radiotherapy. In this context, Lo and colleagues showed that the risk of developing RILD increases from 2.4% to 15.1% with between ALBI grades 1 and 2 [24]. It has therefore been suggested that CPS class A patients with an ALBI grade of 2 should be treated like CPS class B patients. In line with these data, Gkika et al. proposed a shorter follow up period (~1 month after irradiation) for patients with higher baseline ALBI score to account for their higher risk of early and late toxicity, and a more restrictive mean liver dose [29]. This decline may not translate to cholangiocarcinoma. Toesca and colleagues found no correlation between ALBI grade and toxicity following radiotherapy in patients with cholangiocarcinoma (CCA) [25]. The authors suggested that the observed discrepancy between patients with HCC and CCA was based on the smaller patient cohort (*n* = 20 vs. 40 patients), and potential differences in underlying liver fibrosis (15 vs. 65% of patients with fibrosis) between the cohorts. Similarly, ALBI did not have predictive power in patients with CPS class B in two studies [24,27], which again could be due to the small patient sample size but could also reflect a potential inaccuracy of ALBI grade in detecting more severe underlying liver damage. Further prospective studies focusing on these patient subgroups should clarify these inconsistencies.

## 3. Indocyanine Green (ICG) Test

Indocyanine Green (ICG) is an FDA approved tricarbocyanine dye that has been utilized since the 1980s to evaluate hepatic function after several procedures [30,31,32,33]. After intravenous infusion, ICG binds to plasma proteins and subsequently gets excreted into the bile without biotransformation, providing a kinetic estimation of liver function. The speed with which ICG is eliminated depends on several parameters including hepatic blood-flow, allowing for an individualized and sensitive estimation of global liver function [34].

A group at the University of Michigan developed a treatment strategy based on the ICG retention rate 15 min after injection (ICGR15) to individually tailor radiation dose to a patient’s underlying liver function (Table 2, I). In two independent studies (with 48 and 131 patients), they observed that ICGR15 levels mid-treatment, but not prior to irradiation, reflected functional liver reserve after radiotherapy [35,36]. Building upon these findings, they built a toxicity model using ICGR15 levels, which led to the superior prediction of liver toxicity following radiation (AUC = 0.86 vs. 0.75; *p* = 0.04) [37]. Based on these results, the group designed a phase II study where patients were re-assessed for changes in ICGR15 4 weeks after the third of five planned irradiation fractions. If hepatic function was unchanged at this time point, the remaining two fractions were given as initially planned. If liver function had declined, the radiation plan was either adjusted or patients were re-tested after another 4 weeks to allow for more recovery time [38]. The proposed adjustment of radiation dose based on ICGR15 was feasible and led to high local control rates (1y = 99%; 2y = 95%) with low complication rates (increase in CPS 1 or 2 points within 6 months: 14%/7%). Interestingly, a recent study by the same group found that an ALBI-centric model performed equivalently to the ICGR15 model (AUC = 0.79 for both), supporting the use of the less labor-intensive ALBI grade as a biomarker [39]. More importantly, only pre- and mid-treatment levels of ALBI were associated with liver toxicity following radiation. Nevertheless, this approach may be of particular benefit to patients with CPS class B and C liver function, who currently are either excluded from radiotherapy or receive insufficient dosing due to concerns for further hepatic toxicity.

## 4. Hepatocyte Growth Factor (HGF)

Hepatocyte growth factor (HGF) is the ligand to the mesenchymal-epithelial transition factor (MET) tyrosine-kinase receptor and is produced in the liver mainly by activated stromal cells called hepatic stellate cells. The activation of the HGF/MET pathway promotes cell proliferation, survival and invasion [55]. The MET pathway physiologically activates during liver tissue regeneration; however, its constitutive activation in cancer cells can lead to a more aggressive and metastatic phenotype [56,57]. Based on these effects, HGF has been investigated as a biomarker for a variety of cancers, with most of these studies showing a negative correlation between plasma HGF levels and patient survival [58,59,60,61]. For patients with HCC, serum HGF levels were significantly elevated compared to healthy individuals [62].

The specific predictive value of circulating HGF for radiotherapy/SBRT response and RILD remain to be fully characterized [45,62,63] (Table 2, II). In a cohort of 104 patients treated with radiotherapy for liver malignancies, Cuneo and colleagues observed a correlation between pre-treatment HGF levels and overall survival (HGF high vs. low: 14.5 vs. 28.1 months). More interestingly, both pre- and mid-treatment levels of HGF predicted liver toxicity after treatment; however, multivariant analysis showed that mid-treatment HGF levels had the highest predictability [40]. Similarly, in a cohort of HCC and ICC patients treated with proton radiotherapy (*n* = 43), pre-treatment HGF levels were predictive of liver damage after irradiation [41]. Of note, the definition of liver damage considered significant differed between the studies (CTP increase of at least 2 points versus 1 point), as the number of events was small in proton-treated patients. Interestingly, a study by Ng and colleagues did not find a correlation between blood HGF levels and liver toxicity at any time point after radiotherapy [14]. Of note, the reported median HGF levels in these studies differed greatly (1.4 ng/mL (range not reported) [40] vs. 2.31 ng/mL (range 1.037–8 ng/mL) [64] vs. 0.824 ng/mL (range 0.16–11 ng/mL) [45]). In addition, 18 out of 47 patient samples were out of the detection range of the assay [45]. Thus, the observed discrepancy might arise from differences in the patient cohorts investigated or the assay used. Additional insights should be gained from ongoing randomized phase III trial of proton versus photon SBRT in HCC patients (NCT03186898), which will investigate plasma HGF as an integrated biomarker for susceptibility to RILD.

## 5. Cytokines

The release of damage-associated molecular patterns (DAMPs), including cytokines and interleukins, has been reported following irradiation via NF-κB [64,65] (Figure 1). It is well known that irradiation leads to an increase in vessel permeability resulting in both extravasation and recruitment of different leukocyte populations, which in turn creates a pro-inflammatory microenvironment [66]. This inflamed environment induces the release of cytokines, chemokines, and several other factors into the bloodstream, which can be measured by plasma assays such as ELISA. Several studies have evaluated various cytokines and interleukins as potential biomarkers for patients with HCC undergoing radiotherapy (Table 2, III). The following section will focus on markers that have been reported so far.

*CD40 Ligand (CD40L)*, also known as CD154, is mainly expressed on the surface of platelets, endothelial cells and T-cells. It is cleaved by matrix metalloproteases creating the pro-inflammatory molecule soluble CD40 ligand (sCD40L) [67,68]. It has been suggested that patients with cancer have higher levels of sCD40L than patients without cancer [69], and that there may be an association between higher plasma levels of sCD40L and the presence of metastatic disease [70]. For HCC patients undergoing radiotherapy, the potential association between OS and plasma levels of sCD40L has been inconsistent. Ng and colleagues reported that lower levels of sCD40L before treatment were associated with higher risk of death three months after treatment [45]. There was no correlation found in a study by Cuneo et al. (CD40L baseline: HR 0.876; *p*-value = 0.122 and CD40L 1 month: HR 0.907; *p*-value = 0.568). Interestingly, both studies showed that a decrease in sCD40L at mid-treatment was predictive of liver disfunction following radiotherapy. These results suggest that sCD40L may have the potential to mitigate toxicity by facilitating a tailored radiation dose during therapy.

*Interleukine-6 (IL-6)* is a pro-inflammatory cytokine that is known to be elevated in patients with HCC, reflecting chronic inflammation in the liver [71]. Moreover, plasma IL-6 levels have been shown to stratify patients into early vs. late-stage disease and correlate with treatment outcomes after resection or trans-arterial chemoembolization [72,73,74]. With radiotherapy, high levels of circulating IL-6 seem to be associated with overall negative treatment outcomes. Ng and colleagues reported that high levels of soluble IL-6 receptor (sIL-6R) at baseline or mid-treatment, as well as baseline soluble gp160 (sgp130), were correlated with both an increased risk of death three months after treatment and an increased risk of liver toxicity. sIL-6R is the cleaved version of the membrane-bound IL-6R, which plays an important role in trans-signaling of the cytokine IL-6, whereas sgp130 inhibits the signaling of this pathway [75]. In addition, two other groups reported that high circulating IL-6 levels at baseline or mid-treatment were predictive of local failure after radiotherapy (HR 1.15 (95% CI 1.04–1.25) [43] and RR 1.019 (95% CI 1.011–1.028) [76]). Of note, for patients with HCC, this correlation might only be evident in non-treatment-naïve patients [44]. Interestingly, Ajdari et al. found a correlation between the volume of liver radiated with low doses (5Gy, V5) and the risk of liver toxicity, as well as higher plasma IL-6 levels, during treatment [43]. These results indicate that clinically significant increases in circulating IL-6 levels may be detectable even earlier than at mid-treatment. These results are in line with previously published data on increased IL-6 levels following radiation in patients with other cancer types [76,77,78,79,80,81].

*Tumor necrosis factor alpha (TNFα)* is another inflammatory cytokine that can promote tumor progression, survival, and metastasis. TNFα is synthesized as pro-TNFα mainly by NK-cells, T-cells and macrophages and is activated upon enzymatical cleavage. There are two soluble TNFα receptors. TNFα receptor 1 (TNFR1) is ubiquitously expressed. TNFα receptor 2 (TNFR2) is mainly expressed on immune cells and shows a higher affinity for TNFα than TNFR1 [79]. Most biological activity of TNFα is thought to occur via TNFR1 [80]. Although TNFα has been shown to play a major role in development and maintenance of liver inflammation and cirrhosis [81], reports have been inconsistent. Cha and colleagues found no correlation between overall survival and plasma levels of TNFα in patients with HCC undergoing radiotherapy (univariate analysis: *p* = 0.573) [44]. This may be related to the short half-life of TNFα in plasma (ca. 20–70 min) [82] as plasma samples were generated before and after the completion of radiotherapy. Interestingly, two other studies that investigated plasma levels of TNFα receptors over the course of radiotherapy found a correlation with the development of liver toxicity [45,46]. Cousins and colleagues found an association between liver toxicity and TNFR1, while Ng and colleagues found an association with mid-treatment TNFR2 levels. Additionally, Ng and colleagues found that TNFR2 levels were associated with a higher risk of death both at baseline and during treatment. TNFR1 levels were not detectable and thus not evaluated in this study [45]. Further studies are needed to clarify the relationship between the levels of TNFα and TNFα receptors in blood with survival and radiation-induced toxicity in patients undergoing liver radiotherapy.

## 6. Circulating Blood Cells

Radiation-induced lymphopenia (RIL) has been recently identified as a negative biomarker for patients. A recent meta-analysis reported that patients who experienced grade ≥3 lymphopenia after irradiation demonstrated an increased risk of death of 65%, compared to patients who did not [83]. Multiple factors have been reported to influence the risk of developing RIL, including patient age, sex, baseline absolute lymphocyte count (ALC), irradiated tumor volume and fractionation [49,50]. For patients undergoing radiotherapy for liver malignancies, several groups have reported transient decreases in lymphocyte counts [47,48,49,50,51,52] (Table 2, IV). However, the reported duration of RIL differed between studies and ranged between 1 month [48] and up to 1 year [51] following irradiation. Studies have been inconsistent in the type of lymphocyte most impacted by radiotherapy. Gustafson and colleagues observed a decrease in CD3+ T-cells in a more diverse patient cohort including HCC, CCA and patients with liver metastasis [48]. Zhang et al. reported most the prominent change in CD19+ B-cell counts in a cohort of HCC patients [49]. In contrast, Zhuang et al. did not observe any association between CD19+ B-cell, CD3+ T-cell or CD4+ T-cell counts and survival in their patient cohort [51]. Interestingly, the severity of RIL may depend on radiation technique. Zhang and colleagues showed that patients treated with SBRT had the least decline in lymphocytes, compared to patients treated with conventional radiotherapy [49], suggesting that hypofractionation may reduce the risk of developing RIL. Furthermore, it has been proposed that the unique physical properties of proton radiation may reduce the risk and severity of RIL, as protons come to rest within a pre-specified range according to the Bragg peak, reducing the low-dose radiation to healthy liver tissue [84,85,86]. In this context, a recent study comparing lymphocyte levels of patients either undergoing photon or proton radiation for HCC showed that proton therapy led to a higher ALC nadir and significantly longer overall survival (33 vs. 13 months, *p* = 0.002) [54].

Several reports showed a correlation between overall survival and low lymphocyte counts in liver cancer patients, although the specific time points analyzed and parameters evaluated have differed between studies [49,50,51,52]. Of note, markers correlating with overall survival might be specific to cancer histology, as we have previously reported for HCC versus CCA patients [47]. Three studies also reported inverse correlations between the baseline neutrophil-to-lymphocyte ratio (NLR) and treatment outcome (Byun et al.: HR 1.03 (95%CI 1.01–1.06); *p=* 0.016) [48,50,53]. Hsiang and colleagues additionally reported an association between liver toxicity and the maximal change in NLR [liver toxicity rate (delta NLR< vs. >1.9): 7.5% vs. 35.1%].

In summary, the available data suggest that the immune system should be recognized as an organ-at-risk in radiation treatment planning. Investigating the effect of radiation on circulating blood cells and their potential as biomarkers will be crucial in developing strategies for radio-immunotherapy approaches in the future [87].

## 7. Genomic Biomarkers

The advent of next generation sequencing has provided new opportunities for utilizing liquid biopsies for diagnostic purposes [88] (Table 2, V). A promising biomarker in this context is circulating cell-free DNA (cfDNA), which are fragments of 150 to 200 base pairs of double-stranded DNA segments that have a short half-life. Levels of cfDNA been reported to be increased after surgery, in autoimmune disease and other types of inflammation [89]. In a first study of HCC patients undergoing radiotherapy, Park and colleagues observed quantitative changes in cfDNA and stratified patients in low and high cfDNA subgroups [90]. Baseline cfDNA levels correlated with tumor size and stage, as previously reported for other cancer subtypes [91]. Patients in the low cfDNA group following treatment showed improved local control compared to patients with high cfDNA (HR = 2.405 (95% CI 1.059–5.460). The authors hypothesized that continuously elevated cfDNA plasma levels might indicate residual disease or external tumor progression. However, cfDNA levels may be less reliable in patients with acute or chronic inflammatory conditions, as inflammation may also increase cfDNA levels and results must be interpreted accordingly [92].

## 8. Other Soluble Factors

Several other factors have been evaluated by single groups as potential biomarkers in the plasma of patients with both primary and secondary liver cancers undergoing radiotherapy (Table 2, VI). Suh and colleagues observed that plasma levels of vascular endothelial growth factor (VEGF) normalized to platelet count (VEGF/plt) were elevated after radiotherapy [93]. Additionally, higher baseline levels of VEGF/plt were associated with reduced progression-free survival (PFS) and higher risk of outfield intrahepatic failure. Based on these observations, the authors hypothesized that a combination of radiotherapy with anti-angiogenic therapy might have a synergistic effect. A randomized phase III trial (RTOG 1112) is currently investigating the effect of sorafenib alone versus sorafenib in combination with SBRT (NCT01730937), and its results will provide insights into potential synergy. Similarly, Kim and colleagues studied soluble programmed death-ligand 1 (sPD-L1) in the plasma and found that increased levels following radiotherapy correlated with poor survival and the development of metastasis in HCC patients. However, on multivariate analysis, only tumor size remained statistically significant in this study [94]. Radiation-induced upregulation of PD-L1 on cancer cells has been reported both in vitro and in vivo models [94,95,96]. Several trials are currently investigating the combination of radiotherapy with multiple immune checkpoint blockading therapies [96,97]. In an exploratory study, Ng and colleagues observed changes in plasma metabolomic profiles, including the lipid and amino acid metabolism of patients undergoing radiotherapy. Several metabolites were associated with underlying liver function, receipt of radiotherapy and, most interestingly, liver function decline following radiotherapy [98].

Two small trials investigated potential biomarkers in patients undergoing chemoradiation for primary and secondary liver malignancies. One looked at patients undergoing chemoradiation for liver and lung metastases of colorectal cancer (*n* = 35) resulted in a mid-treatment increase in serum ceramide, a pro-apoptotic sphingolipid. Interestingly, high versus low ceramide levels were able to categorize patients into responders and non-responders, respectively [99]. Another promising marker explored in the plasma samples of 10 HCC patients is the inter-alpha inhibitor heavy chain H4 isoform 2 precursor (ITIH4) [100]. Patients were stratified by prognosis, and those in the favorable prognosis group showed significant higher levels of ITIH4 at baseline compared to those with a poor prognosis. Both ceramide and ITIH4 levels need to be evaluated in larger randomized studies to further evaluate their clinical potential.

## 9. Summary and Future Directions

Radiotherapy is a mainstay of treatment for many cancers, and its integration with other therapies holds great promise for liver cancers. Further progress will be facilitated by new strategies for treatment personalization. However, these strategies will depend on the discovery and validation of biomarkers to guide individualized patient management. This limitation is particularly evident in patients with HCC, as they often have underlying liver insufficiency and poorer overall health that often limit management. In recent years, irradiation of the liver has emerged as a valuable treatment option for these patients, but for those with more severe liver dysfunction, its applicability is hampered by concerns of RILD that may result in further impairment of liver function. The availability of predictive and prognostic biomarkers, ones which can also allow for the adjustment of treatment parameters such as dose and field size as treatment progresses, would increase the availability and safety of radiation therapy for a larger group of patients. Blood-based biomarkers may be most practical as they are widely available and can be easily and frequently measured. However, these blood-based biomarkers rely on the assumption that local therapy will result in a measurable systemic effect. Several prognostic and predictive biomarkers have recently been identified to bridge this gap (Table 3). However, most studies have been small and retrospective in nature, with heterogeneous treatment regimens that limit comparability. Interestingly, most markers predictive of liver toxicity following radiotherapy were measured mid-treatment, while the prognostic biomarkers are more typically measured prior to treatment. In addition, most studies have focused on patients with HCC. The unique features of HCC, a highly vascularized tumor often found in the setting of underlying liver fibrosis or cirrhosis, may limit the applicability of these findings to other primary and secondary liver cancers. To increase conclusiveness and validation of these biomarkers, well-powered, prospective randomized controlled trials need to be conducted. Studies are particularly needed for patients with CTP B and C scores, as they currently have limited treatment options and may particularly benefit from radiotherapy. Given the short treatment times with some treatment regimens such as stereotactic body radiation (SBRT), predictive tests with a short turnover time are favored. Alternatively, a test could be performed following half the dose with a planned break, as proposed by colleagues at the University of Michigan Health Services [37,38].

The available data strongly supports the application of the ALBI grade for patients undergoing radiotherapy for liver malignancies. There is compelling evidence that the ALBI grade safely stratifies patients with a CP A score into two distinct predictive and prognostic patient cohorts. Within these cohorts, patients with a CP A score and ALBI grade 1 can be safely treated with radiotherapy within known dose constraints and with minimal risk of serious adverse events. In contrast, patients with a CP A score that show an ALBI grade 2 have a higher risk of developing liver toxicity following irradiation, and should be treated similarly to patients with a CP B score.

## 10. Conclusions

Blood-based diagnostics hold great potential for personalizing radiotherapy in patients with malignant liver disease. Strong evidence supports the notion that patients with CP class A baseline liver function should be further stratified by ALBI score prior to radiation treatment and treated according to their assigned subgroup. Other promising biomarkers are currently being investigated in prospective randomized trials and may provide additional insights into their prognostic or predictive value.

## Figures and Tables

**Figure 1 ijms-23-01926-f001:**
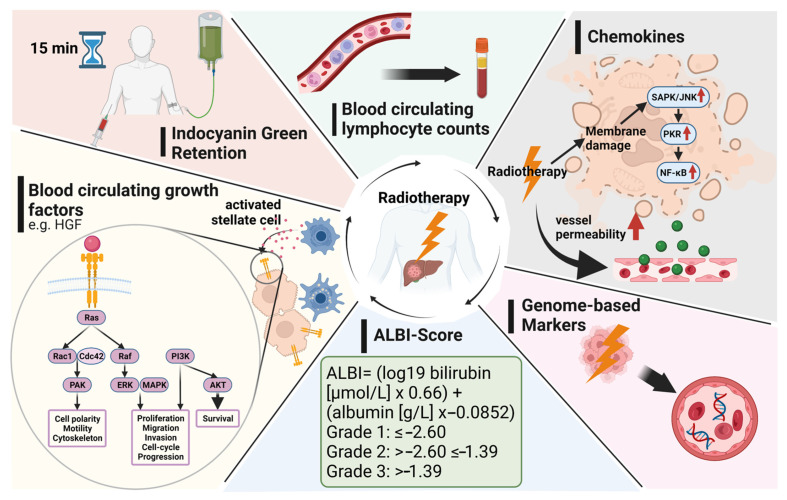
Potential candidates for prognostic and predictive biomarkers in patients with liver malignancies undergoing radiotherapy. HGF, hepatocyte growth factor; RAS, rat sarcoma; Rac1, Ras-related C3 botulinum toxin substrate 1; Cdc42, cell division cycle 42; RAF, rat fibrosarcoma; ERK, extracellular-signal regulated kinase; MAPK, mitogen-activated protein kinase; PI3K, phosphoinositide-3 kinase; AKT, protein kinase B; SAPK/JNK, stress-activated protein kinase/c-Jun NH(2)-terminal kinase; PKR, protein kinase R; NF-kB, nuclear factor kappa-light-chain-enhancer of activated B cells. This figure was created with BioRender.com.

**Table 1 ijms-23-01926-t001:** Albumin–bilirubin (ALBI) grade as biomarker of radiotherapy outcome in liver cancer patients.

Author (Year)	Bio-Marker	Time-Points	Patient #	Cancer Type	Baseline ALBI Score, Median (Range) or Grade 1/2/3 [%]	Underlying Liver Damage (CPS A/B/C/NA [%])	Dose, Median (Range) [Gy]/ Fractionation	Endpoint	Comments
Murray LJ et al. (2018) [23]	ALBI	Pre RT	102	HCC	−2.63 (−3.40 to −1.64)	100/0/0/0	36 (2–54)/6	OS Toxicity	HR (increase in ALBI score per 0.1): 1.09 (95% CI 1.03–1.17) OR (increase in ALBI score per 0.1): 1.51 (95% CI 1.23–1.85)
Lo CH et al. (2017) [24]	ALBI	Pre RT	152	HCC	(−3.67 to −0.84)	78.3/21.7/0/0	45 (25–65)/5 (3–6)	OS Toxicity	HR (increase in ALBI score 2 vs.1): 2.09 (95% CI 1.26–3.46) Pretreatment ALBI Grade: *p* < 0.001
Toesca DAS et al. (2017) [25]	ALBI	pre, 1/3/6/12 months post RT	60	HCC (40/60); CCA (20/60)	5/82.5/12.5	57.5/30/0/12.5	40 (22–50)/5 (1–7)	OS Toxicity	HCC cohort (worsening ALBI score by 0.5 post RT): median OS = 37 vs. 14 months, *p* = 0.0005 CCA cohort (pretreatment ALBI grade): *p* = 0.02 HCC cohort (worsening ALBI score by 0.5 post RT): G3 + HB toxicity= *p* = 0.01; significant decline in hepatic function = *p* = 0.001
Gkika E et al. (2018) [29]	ALBI, inflammation-based index (IBI)	Pre, during, post, 2 months post RT	40	HCC	30/58/12	55/45/0/0	45 (21–66)/3–12	OS Toxicity	Increased OS (lower IBI during treatment): *p* = 0.034 Decreased OS (Higher CRP/AFP): *p* = 0.001 Higher Incidence of acute/late toxicities (Higher ALBI/CPS at baseline): *p* = 0.02/0.001
Jackson WC et al. (2021) [28]	ALBI	Pre RT	151	HCC	25.9/65.7/8.4	66.9/31.3/1.8/0	79.2 (IQR 69.3, 101.7)/3–5	Toxicity	Baseline ALBI: OR 1.8 (95% CI: 1.24–2.62) Change in ALBI: OR 3.07 (95% CI: 1.29–7.32)
Su TS et al. (2019) [27]	ALBI	Pre RT	511	HCC	36.9/58.4/4.7	80.6/18.2/1.2/0	42–43/3–5	OS	Median OS (ALBI grade 1/2/3): 53 vs. 19.5 vs. 6.5 months (*p* < 0.0001)
Ho CH et al. (2018) [26]	ALBI	Pre-RT	174	HCC	−2.39 (−3.61 to −1.41)	100/0/0/0	37.3 (23.3–72)/7 (5–10)	OS	ALBI score: HR = 1.72 (95% CI 1.2–2.48)

ALBI, albumin–bilirubin; CCA, cholangiocarcinoma; CI, confidence interval; CPS, Child–Pugh score; HCC, hepatocellular carcinoma; HR, hazard ratio; IQR, interquartile range; OR, odds ratio; OS, overall survival.

**Table 2 ijms-23-01926-t002:** Candidate biomarkers of radiotherapy outcomes in liver cancer patients.

Author (Year)	Biomarker	Timepoint	Pat. #	Cancer Type [%]	Underlying Liver Damage (CPS A/B/C/NA [%])	Dose [Gy] (Median, Range)/ Fractionation	Endpoint	Comments
(I) Indocyanine Green Retention (ICGR)								
Suresh K et al. (2018) [37]	ICGR after 15 min	Pre and after 3rd fraction, 1/3/6 months post RT	144	HCC	NA	NA/3–5	Toxicity	Inclusion of ICGR15 significantly improves prediction of liver toxicity after irradiation
Feng M et al. (2018) [38]	ICGR after 15 min	Pre and after 3rd fraction	90	HCC (76.7), ICC (4.4), Metastasis (18.9)	NA	49 (23–60)/3 or 5	Phase II Study	High Feasibility of biomarker adapted RT (LC: 1y = 99% (95% CI: 97–100%); 2y = 95% (95% CI: 91–99%)
Stenmark MH et al. (2014) [35]	ICGR after 15 min	Pre, 50–70% of RT dose, 1/2 months post RT	48	HCC (44), ICC (29), Metastasis (27)	92/8/0/0	Different treatment regimes	Toxicity	Both mid-RT ICGR15 and Mean liver dose predicted liver function post RT (*p* < 0.0001)
Lee IJ et al. (2009) [36]	ICGR after 15 min	Pre RT	131	HCC	87/13/0/0	45 +/−16.5/1.5–2.5 Gy/fr	Toxicity	ICGR15 increased after radiotherapy; CPS but not ICGR15 predicted liver toxicity
(II) Hepatocyte Growth Factor (HGF)								
Cuneo KC et al. (2019) [40]	HGF, CD40 Ligand	Pre and after 3rd fraction	104	HCC (84), others (16)	75/22/3/0	28–55/3 or 5; 60/20	OS Toxicity	Pretreatment HGF (High vs. low): 14.5 vs. 27.1 months (*p* = 0.035) Toxicity (Increase in CPS > = 2 points): HGF (baseline/1-month) = OR 6.97 (95% CI 1.05–46.36, *p* value = 0.045)/OR 7.82 (95% CI 1.14–53.6, *p* value 0.036); CD40L (baseline/1-month) = OR 0.47 (95% CI 0.201–1.098, *p* value = 0.081)/OR 0.28 (95% CI 0.086–0.897, *p* value = 0.032)
Hong TS et al. (2018) [41]	Pretreatment HGF	Pre RT	43	HCC (51.2), ICC and others (48.8)	86/14/0/0	58 Gy RBE (15.1–67.5)	OS (2y) PFS (2y) Toxicity	Pretreatment HGF (High vs. low): 14% vs. 69% (*p* = 0.0147) Pretreatment HGF (High vs. low): ns (*p* = 0.348) Low pretreatment HGF: correlation with stable CPS and lower bilirubin (*p* = 0.01)
El Naqa I et al. (2018) [42]	TGFβ1, CCL11, HGF, CD40 Ligand	Pre and after 3rd fraction	192	HCC	NA	SBRT: 49.8 (18.6–60); cf RT: 50.4 (30–90)/3–5	Toxicity	Models to predict liver toxicity after RT were improved by a factor of 1.5 with inclusion of TGFβ1 and Eotaxin
(III) Cytokines and Interleukins								
Ajdari A et al. (2021) [43]	Inflammatory cytokines, gene mutation status, complete blood count	Pre and before 4th fraction	89	Liver metastasis	NA	40 GyE (30–50)/5	OS (2y) LF (1y)	baseline absolute lymphocyte count (High vs. Low): 54% vs. 25% (*p* = 0.0002) Baseline Platelet-to-lymphocyte ratio: HR 1.004 (*p* = 0.0004); Baseline Neutrophile-to-Lymphocyte: HR = 1.32 (*p* = 0.0001) Mutation in KRAS gene (Yes vs. No): 69% vs. 31%; HR 2.92 (95% CI, 1.17 to 7.28, *p* = 0.02) Baseline/mid-treatment interleukin 6: HR 1.15 (95% CI 1.04–1.26, *p* = 0.01)/1.06 (95% CI 1.01–1.13, *p* = 0.01)
Cha H et al. (2017) [44]	IL-1/6/8/10/12, TNF-a	Pre and post RT	51	HCC	96.1/3.9/0/0	50.4 (45–64.8)	OS Infield FFS Outfield-intrahepatic FFS	No correlation between baseline Cytokines and OS baseline serum IL-6 level: *p* < 0.001, RR 1.019 (95%CI 1.011–1.028) Baseline Serum IL-10 level: *p* = 0.026, RR 0.830 (95%CI 0.705–0.978)
Ng SSW et al. (2020) [45]	Soluble cytokine receptors	Pre RT, post 1–2 fractions	47	HCC	81/19/0/0	33(30–54)/6	Risk of early death Toxicity	Lower risk: high baseline level sCD40L = HR 1.8(95% CI 0.27–0.99, *p* = 0.05) Higher risk: high baseline levels sTNFRII = HR 1.93 (95% CI 1.02–3.65, *p* = 0.04); sIL-6r = HR 1.9 (95% CI 1.01–3.57, *p* = 0.05); AFP= HR 2.61 (95% CI 1.03–4.54, *p* = 0.043); sEGFR = HR 2.61 (95% CI 1.32–5.16, *p* = 0.006); sgp130 = HR 2.19 (95% CI 1.13–4.25, *p* = 0.021) ≥2 increase CP score (3 months post RT): increased level sTNFRII (*p* < 0.001); decreased levels of sCD40L (*p* < 0.001)/CXCL1(*p* = 0.01)
Cousins MM et al. (2021) [46]	Soluble TNFa receptor (sTNFR1)	Pre and after 3rd fraction, 1/3/6 months post RT	78	HCC (95), others (5)	NA	18–60/3–5	Toxicity	sTNFR1 (Increase in CPS > = 2 points): baseline= OR 1.62 (*p* = 0.0573); 1 month= OR 2.35 (*p* = 0.0181)
(IV) Circulating Blood cells								
Grassberger C et al. (2018) [47]	Lymphocytes	Pre, Day 8 and Day 15 of RT	43	HCC (51.2), ICC (48.8)	73.7/13.3/0/0	58 RBE/ 15	OS	ICC: baseline CD4 + CD25 + T cells (*p* = 0.003) and CD4 + CD127+ T cells (*p* = 0.01) HCC: mid-treatment fraction of activated CTLs (*p* = 0.007)
Gustafson MP et al. (2017) [48]	Immune cell populations	Pre and post RT, 3 months post RT	10	HCC (50), CCA (10), Metastasis (40)	NA	50–60/5 or 54/3	Changes pre- to post RT	Circulating T cells dropped at the end of RT (2-fold) and recovered within 3 months; CD56br CD16− NK cells dropped 40% after RT and recovered at 3 months
Zhang H et al. (2019) [49]	Lymphocytes	Pre, twice during RT, follow up every 3 months (1st year) then every 6 months	184	HCC	79.3/15.3/0/5.4	75 (50–119) BED/16 (5–35)	OS Toxicity	1/2-year OS (Low vs. high lymphocyte nadir during RT): 56.7% vs. 80.3%; 28.4% vs. 55.7% (*p* < 0.001) Lymphocyte counts declined during RT (*p* < 0.001)
Byun HK et al. (2019) [50]	Lymphocytes	Pre and 3 months post RT	920	HCC	78.2/21.8/0/0	Cf RT: 45–60/20–25; SBRT: 60 or 52/4	OS	Acute severe lymphopenia: HR = 1.40 (95% CI 1.02–1.91), *p* = 0.035 Baseline NLR: HR = 1.03 (95% CI 1.01–1.06), *p* = 0.016
Zhuang Y et al. (2019) [51]	Lymphocytes, TN-Fα	Pre and 10 days, 1/2/3 months post RT, then every 3 months	78	HCC	96.2/3.8/0/0	48 (48–60)/(5–10)	OS	Total peripheral lymphocyte counts post RT < 0.45 × 10^9^/L: HR = 0.14 (95% CI 0.02–0.93), *p* = 0.04 TNFα < 5.5 n/mL: HR = 0.07 (95% CI 0.01-.44), *p* = 0.005
Liu J et al. (2017) [52]	Lymphocytes	Pre and weekly during RT	59	HCC	NA	54 (45–62)/NA	OS	Minimum value of absolute lymphocyte counts (cut-off 300 cells/µL): OR 28.8 (95% CI 27.23–30.37)
Hsiang CW et al. (2021) [53]	Neutrophil -to-Lymphocyte Ratio (NLR)	Pre and 3 months post RT	93	HCC	69.9/30.1/0/0	45 (25–60)/5(4–6)	OS Toxicity	Pre-RT NLR: HR = 1.24 (95% CI 1.12–1.38), *p* < 0.001 Delta NLR: HR = 1.1 (95% CI 1.02–1.18), *p* = 0.011 Liver toxicity rate (delta NLR <vs > 1.9): 7.5% vs. 35.1%
De B et al. (2021) [54]	Lymphocytes	Pre, during, post RT	143	HCC	80/20/0/0	Photon (72%); Proton (28%) 60 (30–100)/15 (3–34)	OS	pre-RT ALC ≤ 0.5: OS (median 7 vs. 20 months, *p* = 0.03); HR = 2.677 (95% CI 1.057–6.779), *p* = 0.039) Post-RT ALC ≤ 0.5: HR = 1.031 (95% CI 1.001–1.062), *p* = 0.043) G3 or higher lymphopenia during RT: OS (median 13 vs. 31 months, *p* < 0.001)
(V) Genomic Markers								
Cuneo KC et al. (2016)c	Micro RNA (miR)	Pre and after 3rd fraction, 1/3/6 months post RT	30	HCC	NA	NA/3–5	Toxicity	Potential correlation with microRNA miR.122.3p, miR.375, miR.217, miR.125a.5p
Park S et al. (2018)	Cell-free DNA	Pre and post RT	55	HCC	88.5/11.5/0/0	SBRT: 60/4; cf RT: 45.6 (45–60)/1.8 Gy/fr (1.8–3) + Ctx	LC Intrahepatic FFS	Post RT (low vs. High cell-free DNA): *p* = 0.041 (SBRT); *p* = 0.046 (cf RT) Post RT cell free DNA = HR 2.405 (95% CI 1.059–5.460)
(VI) Other Soluble Factors								
Dubois N et al. (2016)	Ceramide	D0, D3 (post 2fr), D10 (post 4fr)	35	Liver and lung metastasis (colorectal cancer)	NA	40/4 (Rctx with Irinotecan)	Tumor control (1y)	HR (Ceramide D10): 1.09 (95% CI 1.03–1.17)
Lee EJ et al. (2018)	Inter-alpha Inhibitor H4 (ITIH4)	Pre and post RT	20	HCC	95/0/0/0	45/25 (Rctx with 5FU)	Prognosis	Good Prognosis group (fold change ITIH4 compared to poor prognosis group): 6.1, *p* < 0.05
Kim HJ et al. (2018)	Soluble programmed cell death-ligand 1 (sPD-L1)	Pre and post RT, 1 month after RT	53	HCC	90.6/9.4/0/0	SBRT: 60/4; Cf RT: 45/25 + Ctx	OS (2y) Plasma Level	sPD-L1 (low vs. high): 87.5% vs. 47.7%, *p* = 0.037 Mean sPD-L1 level (pre/post/1 month post RT) [pg/mL]: 6.99 (+/−6.55); 12.93 (+/−8.27); 12.31 (+/−7.72), *p* < 0.001
Suh YG et al. (2014)	Vascular Endothelial Growth Factor (VEGF)	Pre and post RT	50	HCC	96/4/0/0	49 (36–60)/1.8–2.95 Gy/fr	PFS Outfield-intrahepatic recurrence	Worse PFS: high baseline levels of VEGF/Plt = HR 2.22 (95% CI 1.04–4.76, *p* = 0.04) Higher Risk: higher VEGF/Plt levels pre and post RT (*p* = 0.04)
Ng SSW et al. (2020)	Plasma metabolites	Pre RT, post 1–2 fractions	47	HCC	81/19/0/0	33 (30–54)/6	Liver toxicity	Increase in CPS 3 months at least 2 points: increase in serine and alanine

BED, biologically effective dose; ICC, intrahepatic cholangiocarcinoma; Cf, conventional fractionated; CI, confidence interval; CPS, Child–Pugh score; FFS, failure free survival; fr, fraction; HCC, hepatocellular carcinoma; HR, hazard ratio; LC, local control; NA, not available; NK, natural killer cells; OR, odd ratio; OS, overall survival; PFS, progression-free survival; Plt, platelet; RBE, relative biological effectiveness; Rctx, radiochemotherapy; RT, radiotherapy; SBRT, stereotactic body radiation therapy; TGF, transforming growth factor; TNF, tumor necrosis factor.

**Table 3 ijms-23-01926-t003:** Potential prognostic and predictive scores and biomarkers for patients undergoing radiotherapy for liver malignancies.

Potential Prognostic Scores/Biomarkers	Potential Predictive Scores/Biomarkers
ALBI [23,24,25,26,27,29]	ALBI [23,24,25,29,39]
Absolute lymphocyte count [43]	Indocyanin Green Retention [35,36,37]
Hepatocyte growth factor (HGF) [41,40]	HGF [41,40]
CD40 Ligand (CD40L) [45]	sCD40L [45,40]
Platelet-to-lymphocyte ratio [43]	Transforming growth factor (TGF)-β [42]
Neutrophile-to-Lymphocyte ratio [43,50,53]	Neutrophile-to-Lymphocyte ratio [53]
Interleukin 6 (IL-6) [43,44]	Eotaxin [42]
Interleukin 10 (IL-10) [44]	TNF receptor I (TNFR-I) [46]
Tumor Necrosis Factor receptor II [45]	TNFR-II [45]
Circulating lymphocyte counts [47,49,50,51,52]	Circulating lymphocyte counts [49]
Tumor Necrosis Factor (TNF)-α [51]	Micro RNAs [101]
Cell-free DNA [90]	Plasma metabolites [98]
Ceramide [99]	
Programmed cell death ligand 1 (PD-L1) [102]	
Vascular Endothelial Growth Factor (VEGF)/platelets [93]	

## Data Availability

Not applicable.

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
