# Peer review of "Leveraging Blood-Based Diagnostics to Predict Tumor Biology and Extend the Application and Personalization of Radiotherapy in Liver Cancers"

_ijms, 2022, doi:10.3390/ijms23041926_

Round 1
Reviewer 1 Report
Comments for the Authors
- In the introduction, the information about how we obtain and collect the biomarkers is useless.
- Everyone is familiar with Child-Pugh score. All the information about what the child-pugh score is, and how this is calculated are unnecessary
- Child-Pugh score and ALBI are not biomarkers but scores. For that reason the title of your manuscript must be changed
- You describe how some scores such as Child-Pugh score and ALBI affect the prognosis and outcome of patients with HCC undergoing radiation. You must add a paragraph how the classification of patients according to BCLC staging also affects the prognosis and outcome of patients with HCC undergoing liver radiation
- indocyanine green test is not a biomarker. This confirm that the title of your manuscript must be changed
- The paragraph about treatment of HCC with tyrosine-kinase inhibitors or immune check-point inhibitors is out of the scope of this review and must be removed
- In some paragraphs you analyze data regarding the relation of biomarkers with the outcome of patients treated not with radiation but with radiofrequency ablation, TACE or other interventions. These information are out of the scope of the manuscript and must be removed.
- You must explain better which of the biomarkers presented in your paper are the most promising. Which of those biomarkers you believe that they are going to play a significant role in daily clinical practice. It is necessary to give a more clear message
Author Response
See PDF

Reviewer 2 Report
The authors reviewed blood-based (soluble) biomarkers that are potentially useful to predict the efficacy of radiotherapy for hepatic cancer from patient to patient. It is well-written, covering a variety of candidates. It may not be conclusive but informative.
One minor point: It is probably more like an issue for the publisher. Each column in Table 2 is too short to read easily. Please use smaller font size to improve a view at a glance.
Author Response
See PDF

Round 2
Reviewer 1 Report
The new version of the manuscript is better
Grammar and syntax problems regarding the English language have been improved
The new title reflects better the scope of this review
All of my comments have been substantially answered
I have nothing to add